# Effect of Protease Type and Peptide Size on the In Vitro Antioxidant, Antihypertensive and Anti-Diabetic Activities of Eggplant Leaf Protein Hydrolysates

**DOI:** 10.3390/foods10051112

**Published:** 2021-05-18

**Authors:** Akinsola A. Famuwagun, Adeola M. Alashi, Saka O. Gbadamosi, Kehinde A. Taiwo, Durodoluwa Oyedele, Odunayo C. Adebooye, Rotimi E. Aluko

**Affiliations:** 1Department of Food and Human Nutritional Sciences, University of Manitoba, Winnipeg, MB R3T 2N2, Canada; Monisola.Alashi@umanitoba.ca (A.M.A.); sunkanmig@yahoo.com (S.O.G.); Rotimi.Aluko@umanitoba.ca (R.E.A.); 2Department of Food Science & Technology, Obafemi Awolowo University, Ile-Ife 220002, Nigeria; kehindetaiwo3@yahoo.com; 3Department of Soil and Land Resources Management, Faculty of Agriculture, Obafemi Awolowo University, Ile-Ife 220002, Nigeria; d_oyedele@yahoo.com; 4Department of Agronomy, Faculty of Agriculture, Obafemi Awolowo University, Ile-Ife 220002, Nigeria; ocadebooye@daad-alumni.de

**Keywords:** antioxidant, enzyme inhibition, leaf protein, *Solanum macrocarpon*, protein hydrolysate, membrane ultrafiltration, peptide sequence

## Abstract

*Solanum macrocarpon* (eggplant) leaf protein isolate (ELI) was hydrolyzed using four different enzymes to produce hydrolysates from alcalase (AH), chymotrypsin (CH) pepsin (PH) and trypsin (TH). CH had an overall stronger antioxidant property and was separated using ultrafiltration membranes into <1, 1–3 and 3–5 kDa peptide fractions. Gel-permeation chromatography confirmed conversion of the ELI (average of 22 kDa) into protein hydrolysates that contained smaller peptides (<6 kDa). A total of 23 peptides consisting of tri and tetrapeptides were identified from the CH, which is a wider spectrum when compared to seven for AH and four each for TH and PH. CH exhibited stronger scavenging activities against DPPH and hydroxyl radicals. CH and TH exhibited the strongest inhibitions against angiotensin-converting enzyme. In contrast, AH was the strongest inhibitor of α-amylase while AH and PH had strong inhibitory activities against α-glucosidase when compared with other hydrolysates. Ultrafiltration fractionation produced peptides that were stronger (*p* < 0.05) scavengers of DPPH, and hydroxyl radicals, in addition to better metal-chelating and enzyme inhibition agents. The study concluded that the eggplant protein hydrolysates and the UF fractions may find applications in tackling oxidative stress-related diseases and conditions involving excessive activities of the metabolic enzymes.

## 1. Introduction

Over the years, there have been much research with respect to the production of bioactive peptides from plant sources. The choice of plant proteins in the production of novel and natural bioactive peptides may not only be due to their natural sources of supply, affordability and availability, but also because they are made up of cryptides, which are specific amino acid sequences encrypted within the primary structure of proteins with health benefits beyond basic nutritional attributes [1]. The recognition of these bioactive peptides as possible ingredients in functional foods and nutraceuticals development is connected to their perceived desirable health benefits [2]. Some of the benefits of these bioactive plant protein-derived peptides include antidiabetic [3], antioxidant [4], antimicrobial [5] and antihypertensive effects [6]. However, these peptides are not active when they are available within the primary protein structure, but exhibit beneficial health effects when released from the parent protein [6].

One popular method to release the cryptides from the primary protein is enzymatic hydrolysis, which involves the use of food-grade proteases to break down the peptide bonds and release a mixture of peptides, called hydrolysates [7]. Subsequent to the proteolytic treatment is peptide fractionation, which can be conducted with ultrafiltration (UF) separation using membranes. The UF method involves separating the crude protein hydrolysate into fractions according to their molecular size [8]. Based on this principle, peptides from various dietary proteins have been enzymatically released and found to exhibit various desirable health benefits. For example, enzymatic protein hydrolysates from pea seeds, hazelnut and watermelon seeds have been identified to exhibit antioxidant, angiotensin-converting enzyme (ACE) and α-amylase inhibitory activities [9,10,11].

Inhibition of enzymes such as α-glucosidase and α-amylase, which are involved in the increase of blood glucose level, is one of the many approaches to verify the antidiabetic properties of peptides isolated from food sources. α-Glucosidase is located in the brush border, with the functionality of hydrolyzing 1,4-α-glycosidic linkages present in oligosaccharides, then converting them into monosaccharides absorbable from the intestine into the blood stream [2]. α-amylase is a digestive enzyme that catalyzes the hydrolytic breakdown of 1,4-α-glycosidic bonds in polysaccharides, especially starch into products of small molecular weight, that are substrates for α-glucosidase activity. Therefore, inhibition of these two enzymes would lead to a reduction in blood glucose levels, which is one of the ways to manage diabetes mellitus [12]. Several protein hydrolysates from plant sources such as pea seeds [3] and potato tubers [13] have been produced with potentials to inhibit α-glucosidase and α-amylase activities.

ACE plays a major physiological role in the renin–angiotensin system, which regulates mammalian blood pressure. Recently, there has been an increase in the study of plant protein-derived ACE-inhibitory peptides due to evidence of reductions of blood pressure in spontaneously hypertensive rats, which indicates antihypertensive effects comparable to those of commercially available drugs [14,15]. Some of the widely studied antihypertensive protein hydrolysates were produced from plants such as neem seed [11], cotton seed [16], oat [17], peanut [18] and rice [19].

In addition, peptides can also play major roles in the management of oxidative stress, which is responsible for the formation of highly reactive and lethal radicals that could promote the development of chronic metabolic disorders, including cancer, hypertension and coronary heart diseases [20]. The close positive relationship between oxidative stress and chronic diseases such as hypertension and diabetes suggests that the use of antioxidants would be an effective preventive or treatment strategy. Several plant proteins have been used as starting materials in the production of protein hydrolysates and found to demonstrate potential to reduce oxidative stress through their antioxidant activities. Some of the plant proteins include those isolated from alfalfa leaf [21], amaranth leaf [22], oat seed [17] and pea seed [18].

Leaf proteins, the concentrated products of plant leaves, are the most abundant and renewable proteins in nature and can serve as better alternatives to expensive animal and seed-based proteins. Moreover, leaf proteins have become a major source of dietary protein in many developing countries, and could potentially form a major protein source for food applications [23]. African eggplant (*Solanum macrocarpon*) leaf is a leafy vegetable, commonly grown in the southwestern part of Nigeria. It is one of the indigenous underutilized vegetable leaves consumed by resource-poor women in the southwest Nigeria to supplement daily requirements for minerals and vitamins [24]. The dried leaf of *S. macrocarpon* has a protein content of 24.90% [22], which may be used to produe proteins and enzymatic hydrolysates. Until now, various edible leaves such as alfalfa [21], curry [25], amaranth [26] and fluted pumpkin [27], have been used to obtain leaf peptides. However, there is scant information on the use of enzymes for the production of hydrolysates from *S*. *macrocarpon* leaves and their associated bioactivities. This study aimed at determinining the effect of protease type and peptide size on the antioxidant properties and inhibitions of the activities of ACE, α-amylase and α-glucosidase.

## 2. Materials and Methods

### 2.1. Preparation of Leaf Powder

Fresh leaves of *Solanum macrocarpon* were obtained from the Teaching and Research Farm of the Obafemi Awolowo University, Ile-Ife, Nigeria. The leaves were subjected to processing methods including sorting, destalking and rinsing. The rinsed leaves were dried using a Uniscope SM9053 Laboratory Oven made in Singerfried, England at a 55 ± 2 °C drying temperature for 8 h. The dried leaves were milled with a VLC sapphire IS-4930 grinder, made in Edinburgh, England. Chlorophyll and other polyphenolic compounds were removed from the milled powder by making a suspension of the powder in acetone at 10% (*w*/*v*) concentration. Removal of chlorophylls from the dried leaves was enhanced by placing the leaf powder/acetone suspension on a magnetic stirrer with continous stirring for 120 min. The solids were then separated from the solvent with a muslin cloth. The retained solids were re-extracted with acetone, filtered as usual, and the residual acetone was removed by air-drying the solids in a fume hood for 48 h, followed by packing into an air-tight container and storage at −20 °C.

### 2.2. Preparation of Leaf Protein Isolate

Eggplant leaf protein isolate (ELI) was obtained by a modified method of Malomo et al. [28]. A 5% (*w*/*v*) suspension of the eggplant leaf powder was prepared in deionised water. The suspension was heated to 37 °C and the pH adjusted to pH 10.0 using 2 M NaOH with continous stirring for 1 h. The mixture was centrifuged at 3500× *g* for 30 min under a refrigerated condition (4 °C) to separate the insoluble residue from the supernatant. The supernatant was then acidified by adjusting to pH 4.5 with 2 M HCl under continous stirring for 20 min, and the precipitate collected by centrifugation (3500× *g* for 30 min at 4 °C). In order to remove salts that were formed in the process, the precipitate was mixed with distilled water, centrifuged (3500× *g* for 30 min at 4 °C) and the supernatant discarded. The washed precipitate was redispersed in deionized water, freeze-dried and stored as the ELI at −20 °C.

### 2.3. Enzymatic Hydrolysis of ELI 

The ELI was hydrolyzed using four proteases, including pepsin, trypsin, alcalase and chymotrypsin enzymes. These proteases were obtained from Sigma-Aldrich, St. Louis, MO, USA. The protesases were added to the protein isolate (5%, *w*/*v*, in water) at 1:100 (enzyme:substrate) concentration. The hydrolysis conditions used for the proteases included pH 8.0 and 50 °C for alcalase, pH 8 and 37 °C for trypsin, pH 2.0 and 37 °C for pepsin, while pH 8.0, 37 °C was used for chymotrypsin. The pH of the hydrolysis process was maintained with either 1 M NaOH or 1 M HCl. The temperature was stabilized using a thermostat, and the hydrolysis conducted for 4 h. The enzymes were inactivated by heating and holding at 85 °C for 15 min, and the supernatant was collected after centrifugation (9000× *g* for 30 min at 4 °C). Each supernatant was freeze-dried as the respective enzymatic hydrolysate and stored at −20 °C.

#### Measurement of Degree of Hydrolysis (DH)

During enzymatic digestion, aliquots were removed every 30 min for DH determination using the ortho-phthaldehyde (OPA) method described by Charoenphum et al. [29], which was modified as follows. The freshly prepared reagent consisted of 6 mM OPA (dissolved in methanol) and 0.2% (*v*/*v*) 2-mercaptoethanol in 50 mM sodium tetraborate containing 1% (*w*/*v*) sodium dodecyl sulfate. To a 96-well microplate containing hydrolysate or protein isolate, 0.2 mL OPA reagent and 5 µL of standard (gly-gly-gly) were added and the mixture incubated for 100 s at room temperature. The absorbance of the mixture was read at 340 nm using a microplate reader (Multiskan Thermo Fischer Scientific, Waltham, MA, USA) and the degree of hydrolysis was calculated using the Equation (1) below:(1)$$DH\;\left(\%\right)=\frac{{\left(NH_{2}\right)}_{x}-{\left(NH_{2}\right)}_{0}}{{\left(NH_{2}\right)}_{total}-{\left(NH_{2}\right)}_{0}}\times 100$$
where (*NH*_2_)*_x_* is the number of free amino groups at time ‘*x*’ during enzymatic hydrolysis and (*NH*_2_)*_total_* is the total number of amino groups (obtained from 24 h 6 M HCl hydrolysis of ELI). (*NH*_2_)_0_ represents the free amino groups at 0 min (start of enzymatic hydrolysis).

### 2.4. Membrane Ultrafiltration (UF) of Protein Hydrolysates

Relative to the other hydrolysates, the chymotrypsin hydrolysate (CH) exhibited superior enzyme inhibitory and antioxidant activities and, therefore, was fractionated in an Amicon 8400 ultrafiltration stirred cell (Millipore Corp., Billerica, MA, USA) using 1, 3 and 5 kDa MWCO membranes in a sequential manner. First, the CH was passed through the 1 kDa membrane to collect a <1 kDa permeate. Second, the 1 kDa retentate was passed through a 3 kDa membrane and the permeate obtained as the 1–3 kDa fraction. Finally, the retentate from the 3 kDa was passed through a 5 kDa membrane to obtain a 3–5 kDa permeate and the retentate was discarded. The membrane permeates were freeze-dried and stored at −20 °C. The Lowry method [30] was used to determine the protein contents of ELI, protein hydrolysates and UF fractions.

### 2.5. Amino Acid Composition 

Samples were first hydrolyzed at 110 °C with 6 M HCl for 24 h and neutralized with 25% NaOH [31]. Cysteine and methionine were determined after performic acid oxidation [32] while the concentration of tryptophan was measured after alkaline hydrolysis [33]. Amino acid quantification was then performed on an ion-exchange chromatography column (4.6 × 150 mm^2^) with a Sykam Amino Acid Analyzer, Model S2100/S4300 following the procedures outlined by the manufacturer (Skyam GmbH, Eresing, Germany). The amino acids were separated using a gradient consisting of sodium citrate buffers (pH 3.45 and pH 10.85) at 0.45 mL/min flow rate.

### 2.6. Mass Spectrometry and Identification of Peptide Sequences

The method described by Malomo and Aluko [34] was used to identify the peptide sequences of the protein hydrolysates. Formic acid (0.1% *v*/*v*) was used to dilute the sample to a final concentration of 10 ng/µL. The mixture was filtered using 0.2 µm and 10 µL of the filtrate and loaded onto an Absciex QTRAP^®^ 6500 mass spectrometer (Absciex Ltd., Foster City, CA, USA) coupled with an electrospray ionization source. The working conditions of the equipment included 3.5 kV ion spray voltage at 150 °C, and 30 µL/min flow rate for 3 min in the positive ion mode with 2000 Da m/z scan maximum. The peptide sequences were obtained from the published primary sequence of *S. macrocarpon* ribulose*-1,5-*bisphosphate carboxylase-oxygenase (Rubisco) with (±0.025 Da mass tolerance) using the ExPASy Proteomics Server FindPept tool (http://web.expasy.org/findpept/; accessed on 28 February 2021).

### 2.7. Determination of Molecular Weight (MW) Distribution

The molecular weight (MW) distribution of protein hydrolysates was evaluated using the method described by Alashi et al. [35] A 1 mL sample aliquot (5 mg/mL) prepared in phosphate buffer, pH 7.0 containing 0.15 M NaCl was loaded onto a Superdex™ Peptide 10/300 GL column (10 × 300 mm) connected to an AKTA FPLC system (GE Healthcare, Montreal, PQ, Canada) and eluted (0.5 mL/min flow rate) at room temperature using phosphate buffer. The average molecular weight of each peak was estimated by extrapolating the elution volume to a linear plot of log MW versus elution volume of standard proteins (cytochrome C, 12.38 kDa; aprotinin, 6.51 kDa; vitamin B12, 1.85 kDa; glycine, 0.075 kDa).

### 2.8. Determination of Antioxidant Properties

#### 2.8.1. 1,1-Diphenylpicrylhydrazine (DPPH) Radical Scavenging Activity 

The method described by Girgih et al. [36] was used to determine the DPPH radical scavenging activity of the samples. Briefly, sample solutions (0.0156–1.0 mg/mL assay concentrations) were prepared using 0.1 M sodium phosphate buffer at pH 7.0 and containing 1% (*v*/*v*) Triton-X. The DPPH was prepared to a final concentration of 0.1 mM using 95% methanol. To a clean and dry 96-well plate, 0.1 mL of each sample and 0.1 mL of the DPPH solution were pipetted and the plate was subsequantly incubated in the dark for 30 min at room temperature. The buffer (0.1 mL) was used as the blank, while 0.1 mL glutathione (GSH) served as the positive control and the absorbance was determined at 517 nm using a microplate reader. DPPH radical scavenging activity (DRSA) was calculated using the following Equation (2)
(2)$$DPPH\;radical\;scavenging\;activity\;\left(\%\right)=\frac{\left(A1-A2\right)}{A1}\times 100$$
where *A*1 is the absorbance of the blank and *A*2 is the absorbance of the sample. The effective concentration of sample that scavenged 50% of the free radicals (EC_50_) was obtained using a nonlinear regression plot of DRSA (%) versus sample concentration (mg/mL).

#### 2.8.2. Superoxide Radical Scavenging Activity (SRSA)

The method described by Xie et al. [21] was used to determine the SRSA of the protein samples at 0.25–1.5 mg/mL concentrations. Each sample concentration was dissolved in the buffer (50 mM Tris–HCl, pH 8.3 with 1 mM EDTA). To a clear bottom microplate, 80 µL sample solution (or buffer as the blank) and 40 µL of 1.5 mM pyrogallol (dissolved in 10 mM HCl) were also added together in the dark. Changes in the reaction rate were then determined immediately at room temperature for 4 min (1 min intervals) at 420 nm with a microplate reader. The SRSA was calculated using the following Equation (3):(3)$$SRSA=\frac{slope\;of\;blank\;for\;SRSA-slope\;of\;sample\;for\;SRSA\;}{slope\;of\;absorbance\;per\;minute\;of\;blank\;of\;SRSA}\times 100$$

The EC_50_ value of each sample was determined by nonlinear regression from a plot of SRSA (%) activity versus sample concentration (mg/mL).

#### 2.8.3. Hydroxyl Radical Scavenging Activity (HRSA) 

The method described by Ajibola et al. [37] was used to determine the hydroxyl radical scavenging activities of the protein samples using 0.1–4 mg/mL concentrations. The samples and 3 mM 1,10 phenanthroline solutions were prepared using phosphate buffer (0.1 M, pH 7.4). Separately, distilled water was used to prepare 3 mM FeSO_4_ and 0.01% (*v*/*v*) hydrogen peroxide solutions. After incubation for 30 min at room temperature, 50 µL each of sample or GSH, 1,10-phenanthroline, FeSO_4_ and hydrogen peroxide solutions were added together in the microplate followed by incubation at 37 °C for 1 h with constant shaking. For the blank well, 50 µL phosphate buffer was added instead of sample. The color intensity of the reaction mixtures was measured at 536 nm every 10 min for 1 h using a microplate reader. The rate of reaction (∆A/min) was used to calculate the hydroxyl radical scavenging activities as shown in the Equation (4) below
(4)$$HRSA=\frac{\Delta A/\min\;of\;blank-\Delta A/\min\;of\;sample\;\;\;}{\Delta A/\min\;of\;blank}\times 100$$

The EC_50_ values were calculated from a nonlinear regression plot of HRSA (%) activity against sample concentration (mg/mL).

#### 2.8.4. Ferric Reducing Antioxidant Power (FRAP) 

The FRAP of the samples was determined following the method described by Benzie and Strain [38]. Each of 300 mM acetate buffer (pH 3.6), 10 mM of 2,4,6-tri-(2-pyridyl)-1,3,5-triazine and 20 mM of FeCl_3_ were prepared separately and the three solutions mixed together in the ratio 5:1:1 to obtain the FRAP working reagent. The straw-colored working reagent was warmed to 37 °C before use. Each protein sample was prepared to a concentration of 0.167 mg/mL using distilled water and centrifuged to obtain a clear solution. To a 96-clear well plate was added 40 µL of the sample, followed by 200 µL of FRAP working reagent. The absorbance of the mixture was measured immediately at 593 nm using a microplate reader. Iron (II) sulfate heptahydrate (FeSO_4_·7H_2_O) was prepared in a similar manner as the samples, but to concentrations between 0.025–0.25 mM, and the FRAP value of the samples were extrapolated from the FeSO_4_·7H_2_O calibration curve and expressed as Fe^2+^ (mM).

#### 2.8.5. Metal Chelation Activity (MCA) 

The FeCl_2_/ferrozine method of Xie et al. [21] was used to determine the MCA of the samples. Distilled water was used as a blank and also to dissiolve the samples (including GSH) to 1–5 mg/mL concentrations. To different clean and dry test tubes containing 1 mL sample, standard or blank, 50 µL of 2 mM FeCl_2_, 1.85 mL of distilled water and 100 µL of 5 mM ferrozine were added in sequentially. Then, to a clear 96-well plate, 200 µL of the mixture from the reaction tubes were pipetted and the absorbance measured using microplate reader at 562 nm. The MCA was then calculated using the Equation (5) below.
(5)$$Metal\;chelation\;activity\;\left(\%\right)=\frac{\left(Ab-As\right)}{Ab}\times 100$$
where *Ab* and *As* represent absorbance of the blank and sample, respectively. The EC_50_ was calculated by nonlinear regression from a plot of MCA (%) versus sample concentration (mg/mL).

#### 2.8.6. Inhibition of Linoleic Acid Oxidation

The inhibitory properties of the samples against linoleic acid oxidation were evaluated following the method described by He et al. [39] Sodium phosphate buffer (0.1 M, pH 7.0) was used to dissolve the samples to obtain a 0.25 mg/mL assay concentration. A 1 mL aliquot of the sample solution or blank (phosphate buffer) was mixed with 1 mL of linoleic acid (50 mM linoleic acid dissolved in 95% ethanol). The mixture was incubated at 60 °C, and the degree of linoleic acid oxidation measured daily for seven days. Each day, the reaction tube consisted of 100 µL aliquot of the assay mixture in addition to 4.7 mL of 75% (*v*/*v*) ethanol, 100 µL of 30% (*w*/*v*) ammonium thiocyanate and 100 µL of 0.02 M ferric chloride dissolved in 1 M HCl. After the mixture was incubated for 3 min, 200 µL was transferred into a clear-bottom 96-well plate and the absorbance measured at 500 nm using a microplate reader. An increase in absorbance value was used as an index of linoleic acid oxidation.

### 2.9. Enzyme Inhibitory Activities 

#### 2.9.1. Angiotensin-Converting Enzyme (ACE) Inhibitory Activity

The in vitro inhibitory properties of samples against rabbit lung ACE (Sigma-Aldrich, St. Louis, MO, USA) was determined using the method of Udenigwe et al. [40] with *N*-[3-(2-furyl) acryloyl]-Phe-Gly-Gly (FAPPG) as substrate. Briefly, 1 mL of 0.5 mM FAPGG (dissolved in 50 mM Tris–HCl buffer containing 0.3 M NaCl, pH 7.5, and kept at 37 °C) was mixed with 20 μL of ACE (assay activity of 20 mU) and 200 μL of the samples (0.25, 0.50, 0.75, 1.00, 1.25, 1.50, 1.75, 2.00 mg/mL assay concentration) dissolved in Tris–HCl buffer. The decrease in absorbance at 345 nm was measured for 2 min at room temperature. The Tris–HCl buffer was used as the assay blank. ACE activity was expressed as the change in the reaction rate (Δ*A*/min) and inhibitory activity was determined as follows, the Equation (6).
(6)$$ACE\;inhibition\;\left(\%\right)=\frac{\Delta A/\min\left(blank\right)-\Delta A/\min\left(sample\right)}{\Delta A/\min\left(blank\right)}\times 100$$
where (Δ*A*/min) (blank) and (Δ*A*/min) (sample) are ACE activities without and with sample, respectively. The sample inhibitory concentration that reduced ACE activity by 50% (IC_50_) was calculated by nonlinear regression from a plot of ACE activity (%) against sample concentration (mg/mL).

#### 2.9.2. Inhibition of α-Amylase Activity 

The 3,5-dinitrosalicylic acid (DNSA) method of Kwon et al. [41] was used to determine the inhibitory activities of the samples against porcine α-amylase. A 0.02 M phosphate buffer containing 6 mM NaCl (pH 6.9) was used to prepare the sample (500 µg/mL), acarbose (9 µg/mL), starch (1%, *w*/*v*) and α-amylase (1 mg/mL: purchased from Sigma-Aldrich, St. Louis, MO, USA). Acarbose was used at the positive control because it is an approved α-amylase inhibitory drug. To a clean and dry reaction tube, 100 µL of each sample or acarbose was added, followed by the addition of 100 µL α-amylase (200 units/mg). The reaction mixture was preincubated to 25 °C, then 200 µL of starch solution was added. The reaction mixture was further incubated for 10 min at room temperature. To terminate the reactions, the mixture was heated in boiling water for 5 min and then 1 mL DNSA was added, cooled and diluted with 5 volumes of distilled water. Finally, 200 µL of the diluted mixture was pipetted into the 96-well microplate and absorbance (Abs) was measured at 540 nm in a microplate reader. The percentage inhibition was then determined as follows, the Equation (7).
(7)$$\alpha –Amylase\;inhibition\;\left(\%\right)=\frac{\left[Abs\;of\;control-\left(Abs\;of\;sample-Abs\;of\;sample\;blank\right)\right]}{Abs\;of\;control}\times 100$$

#### 2.9.3. Inhibition of α-Glucosidase

The inhibition of α-glucosidase activity by the samples was determined using the yeast enzyme and *p*-nitrophenyl-α-d-glucopyranoside (pNPG) as substrate according to the method of Kim and Byun [42] A 0.1 M phosphate buffer solution (pH 6.9) was used to dissolve the samples (6 mg/mL), acarbose (0.25 mg/mL), which is an approved α-glucosidase inhibitory drug, and α-glucosidase enzyme (1 mg/mL). To a dry and clear 96-well plate, 50 µL of the enzyme was added, followed by 100 µL of the sample (or acarbose), and the plate was subsequantly incubated for 20 min at 37 °C. A 100 µL aliquot of 5 mM pNPG, which was dissolved in 0.1 M phosphate buffer at pH 6.9, was then added. The mixture was incubated for 10 min at 37 °C and the absorbance (Abs) was measured with a microplate reader at 405 nm for 30 min at 1 min intervals. The percentage inhibitory activity was calculated using the Equation (8).
(8)$$\alpha -Glucosidase\;inhibition\;\left(\%\right)=\frac{\left(Abs\;of\;blank-Abs\;of\;sample\right)}{absorbance\;of\;blank}\times 100$$

### 2.10. Statistical Analyses

The experiments were carried out in triplicate determinations. Data were subjected to analysis of variance using the statistical package for social sciences software (SPSS), version 18. The statistical significance of differences between mean values of the data were determined at *p* < 0.05 level using Duncan Multiple Range Test [37].

## 3. Results

### 3.1. Degree of Hydrolysis (DH) and Protein Content 

The highest protein content was obtained in PH (81.31%) followed by AH (80.55%), CH (77.91%) and TH (72.22%); these values were significantly (*p* < 0.05) different from one another. Protein contents of the CH membrane fractions were between 18.77 and 43.33%. The highest MW fraction (3–5 kDa), had the most protein content (43.33%), followed by 1–3 kDa fraction (38.26%), and then <1 kDa (18.77%). The DH values for AH, CH, PH and TH at 30 min intervals for 4-h are shown in Figure 1. There was rapid hydrolysis of the protein in the first 0.5 h, leading to 12.80–14.15% DH for the hydrolysates. DH increased steadily for all the samples until the 2.5 h mark when the values increased sharply for AH and CH but not TH and PH. At the end of protein hydrolysis, alcalase and chymotrypsin produced the highest DH while pepsin generated the lowest value.

### 3.2. Amino Acid Composition

The amino acid composition of eggplant leaf protein isolate and resulting protein hydrolysates are shown in Table 1. Hydrolysis resulted in greater amounts of acidic amino acids over the precursor protein isolate (ELI). The hydrolysates had greater amounts of branched-chain amino acids and negatively charged amino acids compared to the parent protein. In contrast, there were lower levels of other types of amino acids such essential (EAA), hydrophobic (HAA), positively-charged (PCAA) and sulfur-containing (SCAA), in the hydrolysates when compared to ELI.

### 3.3. Peptide Sequencing

Table 2 shows the amino acid sequences of identified peptides and their location within the eggplant leaf Rubisco primary structure. The results show that 7, 23 and 4 peptides sequences were identified for AH, CH, PH/TH, respectively. Out of the 23 peptides identified in CH, 16 were tripeptides while seven were tetrapeptides. The AH also consists of short-chain peptides with of 2–6 amino acids and is the only hydrolysate with dipeptides. Interestingly, PH and TH, which had lower DH also had the lowest number and longest length of identifiable peptides when compared to AH and CH. The trypsin and pepsin hydrolysates shared two (GFK and HAGT) similar peptides but generally longer peptides were present in the TH.

### 3.4. Molecular Weight Distribution

The gel permeation chromatographic results of eggplant leaf protein hydrolysates are shown in Figure 2. The chain length of the protein hydrolysates ranged between 0.85 to 5.81 kDa, which is lower than the 22.20 kDa for unhydrolyzed ELI. Among the protein hydrolysates, PH had a more uniform size distribution, with the most abundant peptides (peak height) estimated to be 5.8 kDa. This was followed by other peak heights such as TH (2.6 kDa), CH (1.13 kDa) and AH (0.85 kDa).

### 3.5. Radical Scavenging Properties

As shown in Table 3, the DRSA EC_50_ values of the protein hydrolysates (0.23–0.35 mg/mL) were lower than the 1.21 mg/mL for the unhydrolyzed protein, suggesting an improvement when compared to the parent protein. The EC_50_ values obtained for the CH and PH were not significantly different (*p* < 0.05) when compared with the values reported for glutathione, which suggests strong potency of the hydrolysate. In contrast to the protein hydrolyates and GSH, the UF membrane fractions exhibited stronger DRSA potency, with the <1 kDa peptides exhibiting significantly lowest EC_50_ value. Table 3 also shows that the SRSA EC_50_ values for the protein hydrolysates ranged between 0.91 and 0.96 mg/mL, which were not significantly (*p* > 0.05) different from the 0.98 mg/mL for the precursor isolate. The precursor isolate and protein hydrolysates were however, less potent against superoxide radical scavengers when compared to GSH (EC_50_ of 0.79 mg/mL). UF fractionation of CH led to reduced SRSA potency (higher EC_50_ values) of the resultant peptide fractions. The smaller size peptides <3 kDa were more potent superoxide radical scavengers (lower EC_50_ values) than the bigger 3–5 kDa peptides. With the exception of PH, the HRSA of the hydrolysate was significantly (*p* < 0.05) better than that of GSH (Table 3). The CH exhibited strongest HRSA when compared with other hydrolysates and the protein isolate. The membrane-separated fractions of CH demonstrated stronger HRSA when compared with the unfractionated hydrolysate, but there was no observed effect of peptide size.

### 3.6. Metal Chelation Activity (MCA) and Ferric Iron Reducing Antioxidant Power

The CH exhibited significantly (*p* < 0.05) stronger MCA because it had the lowest EC_50_ value (4.00 mg/mL) when compared with other hydrolysates, ELI and GSH (Table 3). The GSH had poor MCA when compared to the protein isolate and hydrolysates. The UF fractions had lower MCA EC_50_ values and indicate that they are better metal chelators than the unfractionated hydrolysate. The results also showed weaker MCA (higher EC_50_ values) as the MW of the fractionated peptides increased from <1 kDa to 3–5 kDa. Figure 3 shows that protein hydrolysates had higher FRAP values (0.10–0.15 mM Fe^2+^), which indicate better ability to donate electrons than the protein isolate (0.02 mM Fe^2+^). There was no significant difference (*p* > 0.05) in the FRAP value of CH and PH, but the two hydrolysates had significantly (*p* < 0.05) higher values (0.15 mM Fe^2+^) when compared to AH and TH. The UF-separated peptides showed greater reducing ability than the parent unfractionated CH with the smallest size peptides (<1 kDa) being the most potent.

### 3.7. Lipid Peroxidation Inhibitory Activity

The potential of the protein isolate and hydrolysates to inhibit lipid oxidation is shown in Figure 4. The absorbance values of the control (absence of an antioxidant) increased gradually from the start of the process and attained the peak value (0.88) on the second day of incubation. The values however decreased progressively for the rest of the incubation period. The absorbance values of GSH, protein isolate and AH also increased at the onset of the experiment and peak values were attained for GSH (0.35) and AH (0.17) on day one, while it was day 2 (0.25) for ELI. The CH, TH and PH exhibited stronger inhibition of linoleic acid oxidation based on their lower absorbance values throughout the incubation period. 

### 3.8. Enzyme Inhibitory Properties

Table 3 shows that the ACE-inhibitory activity of CH and TH was significantly (*p* < 0.05) stronger (lower IC_50_ value of ~2.11 mg/mL) when compared to PH and AH with 2.38 and 2.58 mg/mL, respectively. However, all the protein hydrolysates displayed significantly stronger ACE inhibition than the unhydrolyzed protein (ELI). The fractionated CH peptides exhibited greater ACE-inhibitory potency with lower 1C_50_ values than the unfractionated CH hydrolysate.

The inhibitory activities of protein isolate and hydrolysates against α-amylase enzyme is shown in Figure 5A. The values for the inhibitory activities of the ELI and AH were not significantly (*p* > 0.05) different from one another. However, the values obtained for ELI and AH showed that they possessed stronger inhibitory activities when compared to CH, PH and TH. The α-amylase inhibitory properties of the UF peptide fractions were stronger than the parent CH, based on the percentage inhibitions (52.71–58.73% vs. 34.89%) as shown in Figure 5B. The percentage inhibition of the protein isolate (40.46%) and hydrolysates (35.51–38.87%) and the fractions (52.71–58.74%) were lower when compared to that of acarbose (88.99%).

As shown in Figure 6A, the inhibition of α-glucosidase enzyme by the protein hydrolysates ranged between 43.19 and 48.81%, which were significantly (*p* < 0.05) higher when compared to ELI (39.71%). The CH peptide fractions exhibited higher inhibitions against α-glucosidase when compared to the unfractionated CH (Figure 6B). The smallest size peptide fraction (<1 kDa) was a stronger inhibitor of α-glucosidase in comparison to the larger (1–3 and 3–5 kDa) peptides. However, the protein hydrolysates and peptide fractions were weaker α-glucosidase inhibitors than the standard acarbose.

## 4. Discussion

Protein hydrolysis release peptides into the medium, which can then be collected as the soluble part of the reaction mixture after centrifugation and labeled as hydrolysates [6]. However, the hydrolysates can also contain other soluble molecules such as sugars, phenolic acids and free amino acids. Therefore, the lower protein contents of CH and TH suggest the presence of higher levels of nonpeptide molecules when compared to AH and PH. This was especially the case for the TH, which had the lowest protein content, even though it had the highest DH (a measure of peptide release). The low protein contents of the <1 kDa fraction when compared to the high molecular weight fractions (>1 kDa) may be attributed to the formation of salts during the digestion process, whereby pH adjustment was carried out to optimize enzyme activity. Paiva et al. [43] reported 33.60% and 43.40% for 1–3 kDa and >3 kDa fractions of *Fucus spiralis* L. protein hydrolysate fractions, which are similar to data obtained in this work.

The DH is a measure of rate of proteolysis and reflects the number of peptide bonds broken within a protein. Therefore, the initial sharp increase in DH within the first 30 min indicates rapid rate of proteolysis, which may be attributed to easy degradation of peptide bonds in the ELI. The pattern obtained in this work is similar to the DH of cotton seed hydrolysate where rapid rate of reaction was also observed in the first 20 min of digestion [16]. After the first 30 min, the DH of PH increased only slightly with a maximum of 19.69% at 4 h, which suggests reduced number of susceptible peptide bonds as hydrolysis progressed. In contrast, the DH for the other hydrolysates (AH, CH and TH) had similar increases up to 2.5 h, which were more than the rate observed for PH. After 2.5 h, the DH of AH was almost constant for the rest of the hydrolysis period, with a maximum of 24.17% at 4 h. The results suggest limited number of peptide bonds susceptible to alcalase after the initial 2.5 burst of hydrolysis. Flattening of the DH curves for all the hydrolysates towards the end of hydrolysis period may be attributed to reductions in the number of susceptible peptide bonds or feedback inhibition, whereby the peptides block the active site of the enzymes to reduce substrate access. Differences in the DH after the 4-h process may be attributed to different proteolytic specificities of the proteases used [44]. However, the higher DH for trypsin and chymotrypsin suggest that the two enzymes were less affected by the negative peptide feedback inhibition but, more importantly, reflects the presence of more susceptible peptide bonds when compared to pepsin and alcalase. For trypsin and pepsin hydrolyses, the achieved DH were higher when compared to the 26.9 and 9.0%, respectively, that were reported for cowpea hydrolysates [45] The variation may be due to differences in the source of materials (leaf vs. seed proteins), which is consistent with a previous suggestion that differences in the type of substrate have great influence on the DH [46].

Amino acid composition, structures and their hydrophobicity have been shown to facilitate access of peptides to free radicals and scavenge them as a result of the presence of some amino acids with hydrogen or electron donating abilities [47]. Therefore, changes in amino acid contents could influence bioactive properties of proteins and their hydrolysates. The higher levels of acidic amino acids and others such as alanine, valine, leucine and isoleucine in the protein hydrolysates when compared to ELI was similar to an earlier published report on pumpkin seed hydrolysates [48] The hydrolysates had greater amounts of branched-chain amino acids and negatively charged amino acids in comparison to the parent protein, which agreed with the pattern reported for moringa leaf protein hydrolysates [49]. Variations in the amino acid composition of the hydrolysates reflect the differences in proteolytic specificity of the enzymes during digestion. The presence of some amino acids such as tyrosine, histidine, lysine and methionine is important because they have been found to play crucial roles in improving the antioxidant properties of peptides [48] Similarly, the presence of aromatic amino acids such as histidine and tryptophan with large side groups is important for peptide functionality because these residues have been found to enhance antioxidant activities as a result of their hydrogen donating ability [49]. The availability of hydrophobic amino acids may have promoted the antioxidant properties because these residues enhance solubility of peptides in nonpolar environments, which improves access to and interactions with free radicals [50]. Therefore, the hydrolysates have several groups of amino acids with implications for enhanced bioactivity when compared to the unhydrolyzed protein (ELI).

In addition to amino acid content, the arrangement and length of peptide chains can also influence bioactive properties. Out of the 23 peptides identified in CH, 16 were tripeptides while seven were tetrapeptides; these small peptide sizes are consistent with the high DH. The AH also consisted of short-chain peptides with a range of 2–6 amino acids, which is consistent with the random proteolytic activity. Interestingly, PH and TH had the lowest number and longest length of identifiable peptides, which reflects the lower DH when compared to AH and CH. A previous study suggested that low molecular weight peptides have greater chance to pass through the gastrointestinal tracts faster than the peptides with bigger molecular weights, which could enhance bioactive effects [34]. Based on this principle, the small peptides identified in CH may have stronger bioactive impacts if ingested when compared to the other hydrolysates in this study. The TH and PH shared two (GFK and HAGT) similar peptides, suggesting possible comparison of the catalytic activities towards ELI by these two enzymes. The bioactivities of some of the identified peptides were documented and found on BIOPEP-UWM database of bioactive peptides. For instance, the dipeptides identified in alcalase hydrolysate (MS, AF and FA) were found to exhibit antioxidant, ACE-inhibitory and dipeptidyl peptidase IV-inhibitory activities. In fact, MS and FA peptides were identified to be present in egg white and bean protein hydrolysates, respectively (BIOPEP-UWM database) while AF was found to be an ACE-inhibitor in Sorghum Kafirin hydrolysate [51]. Some of the identified peptides in the CH have been found to exhibit various bioactive properties such as ACE-inhibitory, antioxidant, α-amylase and α-glucosidase inhibitory activities as reported in the BIOPEP-UWM database. In the same vein, the two peptides common to PH and TH (GFK and HAGT) have been reported to exhibit α-amylase and antimicrobial potentials, respectively.

Molecular weight is seen as a measure of the extent of protein hydrolysis, which correlates with the bioactivity of protein hydrolysates. The shorter chain lengths of the hydrolysates confirmed fragmentation of the parent protein during enzyme hydrolysis. The greater efficiency of alcalase enzyme to degrade ELI to averagely smaller chain length may reflect its broad proteolytic specificity [48]. This is reflected in the identified peptide sequences, whereby only the AH contained dipeptides. Earlier studies also reported that alcalase was most potent protease in producing smaller size peptides during hydrolysis of rice dreg [51] and pea [3] proteins.

According to Xie et al. [21], the DPPH radical scavenging is an assay that is based on reduction of the absorbance of a methanolic DPPH solution at 517 nm in the presence of a proton-donating substance [52], because of the formation of diamagnetic molecule by accepting electrons from an antioxidant-rich compound. Similar to the results from this work, a previous report on moringa seed protein also indicated stronger DRSA of protein hydrolysates compared to the parent protein isolate [53]. Among the hydrolysates, the strong DRSA of CH is consistent with it having the highest level of sulfur-containing amino acids (SCAA) and high content of negatively charged amino acidsa (NCAA). The abundance of NCAA coupled with the proton-donating ability of the sulfhydryl (SH) of SCAA could have enhanced the ability of CH to quench DPPH radicals more efficiently than the other hydrolysates. This is consistent with the role of cysteine as a major factor in the antioxidant potency of GSH. The results are similar to previously reported data showing that SCAAs are important for the strong DRSA of peptides [54,55]. The stronger DRSA of CH may be related to the presence of a higher number and wider variety of short-chain peptides (tri- and tetrapeptides) when compared to the longer size peptides and narrower structural features of the other hydrolysates, as shown in Table 2. In contrast to the protein hydrolyates and GSH, the UF peptide fractions exhibited stronger DRSA potency. The results suggest that fractionation reduced existing antagonistic interactions in the protein hydrolysates, which contributed to stronger peptide-DPPH interactions. The results are similar to data reported in earlier published works on freshwater muscle hydrolysate fractions [56] and perilla seed hydrolysate fractions [57], where the UF peptide fractions exhibited stronger radical scavenging activities than the intact hydrolysate.

The superoxide radical (O_2_^•^) is believed to be a weak radical because it cannot initiate lipid peroxidation without the involvement of other free radicals. Preventing or scavenging this radical is necessary because it is a precursor to other more toxic radicals, such as hydroxyl radicals and hydrogen peroxide [58]. Therefore, quenching superoxide radicals could translate into reduction in the risks of attacks from other dangerous radicals on DNA and proteins in the body. The reduction in the SRSA of the fractions may be attributed to increase in the peptide-peptide antagonistic interactions after membrane separation. Past findings on freshwater muscle protein by Dong et al. [56] also reported lower superoxide scavenging activities for separated peptide fractions than the parent hydrolysate.

The hydroxyl radical is an important species of free radical due to its high oxidizing power and the destructive effects towards the body’s vital biopolymers such as DNA and hemoglobin. Therefore, scavenging the hydroxyl radical is a preventive step that supports normal functioning of the human body and prevention of degenerative diseases [55]. CH exhibited the strongest HRSA, which could also be due to the higher levels of SCAA combined with NCAA when compared with other hydrolysates and the protein isolate. The presence of a wider variety of peptides may also have contributed to the stronger HRSA of CH. The results also suggest a greater synergistic effect of short-chain peptides may have enhanced the stronger HRSA of CH, AH and TH when compared to PH and ELI. However, synergy of the short peptides may have been negated in the pepsin hydrolysate, resulting in the weaker scavenging against the hydroxyl radical. 

Iron is involved in various oxidative reactions, leading to the production of deleterious radicals such as the Haber-Weiss and Fenton reactions that have been linked to oxidative stress in humans [59]. Therefore, iron sequestration could prevent the destructive interactions with nutrients. The CH had superior MCA, which may reflect a combined effects of a wide spectrum of short-chain peptides and high levels of SCAA, NCAA and histidine. The short chain length enhances efficient interactions with the iron molecule, while the various amino acid groups provide the required electronic interactions necessary for chelation. In particular, the imidazole group of histidine has been suggested to be involved in hydrogen atom transfer and single electron transfer reactions [60]. The lower MCA EC_50_ values of UF fractions of CH indicate they are better metal chelators than the unfractionated CH. The strong MCA of the separated fractions suggest reductions in antagonistic peptide-peptide interactions, which promoted stronger peptide-iron interactions than when the peptides were present in the hydrolysate. The large amounts of peptides in the CH could have enhanced the antagonistic interactions and led to reduced interactions with iron, and hence lower MCA compared to the separated peptide fractions. Similar findings have been reported for rapeseed [39] and hempseed [35], where the membrane-separated peptide fractions exhibited better MCA than the unfractionated protein hydrolysates. The results also showed weaker MCA as MW of the fractionated peptides increased, which supports the suggestion that short-chain peptides have a better ability to donate electrons more readily than long chains [61].

The capacity of peptides to reduce the ferric ion is used to evaluate reducing power of food proteins and the ability to convert the more reactive ferric to the less reactive ferrous form [62]. The FRAP of protein hydrolysates may be due to the fact that protein hydrolysis yields peptide products with a greater number of electron-donating groups, such as amino and carboxyl groups, than the original protein. Moreover, the low MW of the peptides present in the hydrolysates may enhance the effective release of electrons for the reduction reaction when compared to the protein isolate [56]. The high reducing ability of the fractions may have been favored by virtue of their smaller sizes, which could have enhanced peptide interactions with Fe^3+^ when compared to CH, which contained longer peptides. This is consistent with the higher FRAP value of the <1 kDa when compared to the fractions that contained longer (>1 kDa) peptides, and is similar to the previous data reported for fractionated freshwater muscle peptides [56].

Lipids that contain unsaturated fatty acids with more than one double bond are mostly prone to attacks by metal ions and free radicals, which result in lipid peroxidation (LP). Some of the peroxidized lipid compounds (especially hydroperoxides) are toxic, and their accumulation in cells has been attributed to a number of diseases and clinical conditions such as diabetes, Parkinson disease and Alzheimer’s disease [63]. Inhibiting the formation of these LP by-products by antioxidant agents would help to truncate the damaging effects of the free radicals. The initial sharp increases in absorbance of the control sample within the first two days reflect rapid peroxidation of the linoleic acid. However, decreases in the absorbance values of the control after the second day suggest breakdown of the lipid peroxides and the formation of secondary oxidation products [64]. The strong inhibitory potentials of CH, PH and TH against linoleic acid oxidation may be attributed to the ability of the amino acids of peptides within the hydrolysates to donate electrons to the lipid phase throughout the incubation period. The results showing higher absorbance for the AH system suggest that the presence of dipeptides in AH may not be as desirable as the longer chains present in CH, PH and TH with regard to efficacy of inhibiting the early LP phase. In addition, AH had the weakest metal binding ability (Table 3), which reduces effectiveness to prevent metal-catalyzed LP when compared to the other hydrolysates that have stronger MCA. Consistent with results from this work, the inhibition of primary and secondary oxidation products has also been reported for other natural peptide products such as potato, perilla seed, soy protein and pigeon pea protein hydrolysates [4,57,65,66].

ACE plays a crucial role in blood pressure regulation because it catalyses conversion of the inactive angiotensin-I into the potent vasoconstricting angiotensin-II, and also inactivates the potent vasodilator, bradykinin [67]. Inhibition of ACE activity may help in blood pressure modulation, which is one of the ways to manage hypertension. The strong inhibitions of ACE activity by protein hydrolysates may be explained on the basis of the availability of smaller MW peptides (0.85–5.81 kDa) in the hydrolysates when compared to the protein isolate (22.20 kDa) as shown in Figure 2 [68]. This is because the small size of the peptides coupled with the presence of multiple charges could enhance interactions with the ACE protein, especially the ability to penetrate and block the active site. Variations in the amino acid composition and sequences of peptides in the hydrolysates may explain the differences in their ACE-inhibitory activities. The stronger ACE-inhibitory activity of the peptide fractions suggests that the UF separation of CH led to reduced antagonistic peptide-peptide interactions, while enhancing peptide-ACE interactions. The IC_50_ values (0.81–0.84 mg/mL) obtained for the peptide fractions are lower when compared with 1.20–1.36 mg/mL for chickpea [69] and 0.90–1.2 mg/mL for buckwheat protein fractions [70], but higher than that of egg white protein hydrolysate [71]. The variations in the IC_50_ inhibitory activities may be related to differences in the source of the protein substrate used for enzyme hydrolysis (seed vs. leaf and plant vs. animal).

α-amylase is a digestive enzyme that catalyzes the hydrolytic breakdown of polysaccharides into maltose, glucose and maltriose, among other compounds. Being one of the major enzymes of carbohydrate digestion in the body, its inhibition would lead to a decrease in blood glucose levels. The inhibitory values (34.51–38.87%) obtained for the protein hydrolysates indicate lower potency against α-amylase when compared to the values reported for cowpea (50–80%) [45] and watermelon (60–77%) [11] seed protein hydrolysates. The differences in the inhibitory activities may be attributable to substrate variation (leaf vs. seed). The α-amylase inhibitory property of the UF peptide fractions (52.71–58.73%) was stronger than the parent CH (34.89%) as shown in Figure 5B. The higher α-amylase inhibition may be attributed to the greater effectiveness of short-chain peptides in releasing electrons, which enhances interactions with α-amylase to impede substrate catalysis. The positive influence of small peptide size is further supported by higher α-amylase inhibition by the <1 kDa fraction when compared to the >1 kDa sizes.

α-Glucosidase hydrolyses the α-1,4 glycosidic linkages present in oligosaccharides and converts them to monosaccharides that can be absorbed into the blood stream. When the activities of the enzyme are retarded, the rate of glucose absorption and level in the blood stream are reduced, which is one of the effective ways to manage diabetes. The stronger α-glucosidase inhibition by the protein hydrolysates suggest that enzymatic hydrolysis led to the production of peptides with greater ability to attack the active or nonactive sites of the enzyme when compared to the unhydrolyzed ELI. The CH peptide fractions exhibited greater inhibition against α-glucosidase when compared to the unfractionated CH, which suggests increased peptide-enzyme interactions as a result of peptide fractionation. Peptide size influenced the α-glucosidase inhibitory activity with the smallest fraction being more effective than bigger peptides. The results indicate that the <1 kDa peptides have better penetration of the enzyme active site or stronger interactions with nonactive sites to interfere with catalytic action and reduce the reaction rate. A previous study also showed that the <1 kDa peptide fraction of perilla seed protein hydrolysate was the most effective α-glucosidase inhibitor when compared to larger peptides [59]. Other reports have also shown strong α-glucosidase inhibitory property of low MW peptides (3–5 amino acid residues) isolated from *Aspergillus oryzae* [72] and Chinese giant protein salamander [73].

## 5. Conclusions

The peptide composition and in vitro bioactive properties of protein hydrolysates produced from eggplant leaf protein isolate were dependent on the enzyme type, which indicates differences in peptide bond specificity. Estimated MW distribution showed that alcalase-catalyzed hydrolysate contained larger numbers of small molecular weight compounds, especially dipeptides, which was attributed to broad proteolytic specificity. However, the DH showed that trypsin and chymotrypsin exhibited higher peptide cleavages when compared to alcalase and pepsin after 4 h of protein hydrolysis. The superior antioxidant properties of CH were attributed to the presence of a wider spectrum of peptides. The strong antioxidant and enzyme-inhibitory potential of CH were further strengthened upon fractionation into short-chain peptides. The results suggest that eggplant leaf protein hydrolysates, especially the CH and its peptide, may find applications in tackling oxidative stress-related diseases and conditions involving excessive activities of the metabolic enzymes.

## Figures and Tables

**Figure 1 foods-10-01112-f001:**
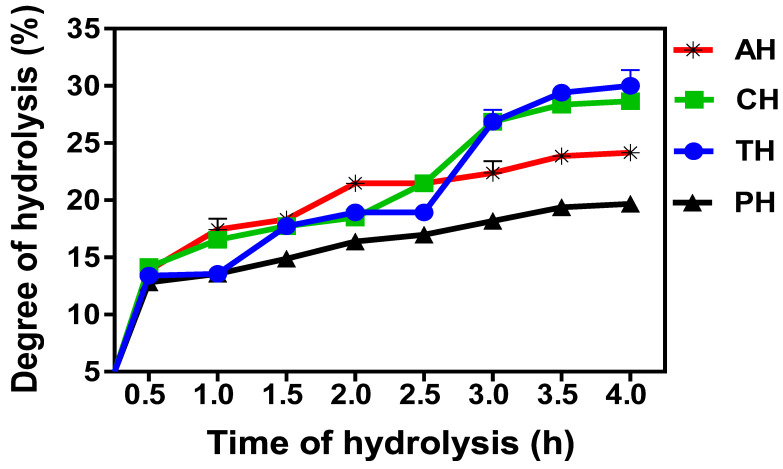
Degree of hydrolysis of alcalase (AH), chymotrypsin (CH), pepsin (PH) and trypsin (TH) eggplant leaf protein hydrolysates.

**Figure 2 foods-10-01112-f002:**
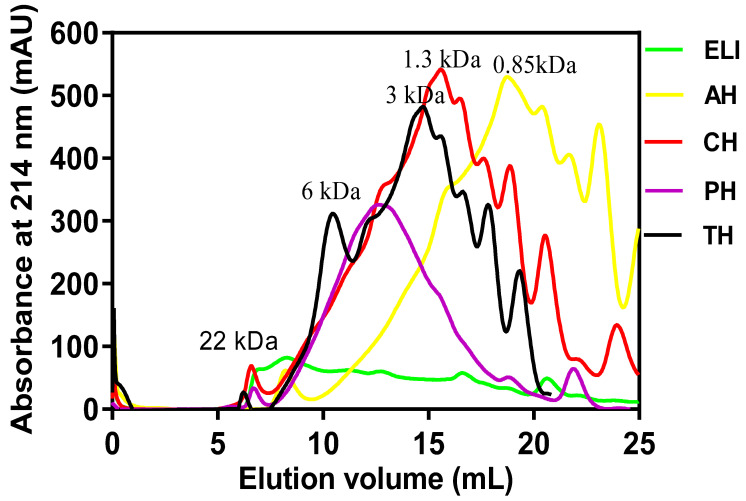
Comparative gel-permeation chromatograms of eggplant leaf protein isolate (ELI) and enzymatic protein hydrolysates: alcalase (AH), chymotrypsin (CH), pepsin (PH) and trypsin (TH). Inserted values indicate estimated molecular weights.

**Figure 3 foods-10-01112-f003:**
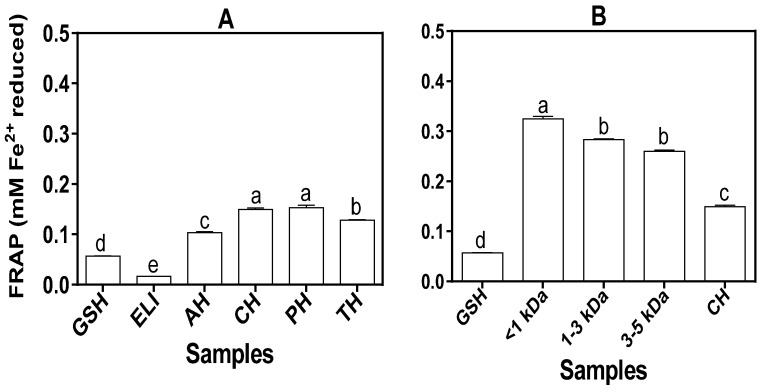
(**A**). Ferric reducing antioxidant power of glutathione (GSH), eggplant leaf protein isolate (ELI) and hydrolysates: alcalase (AH), chymotrypsin (CH), pepsin (PH) and trypsin (TH). Hydrolysates. (**B**). FRAP of chymotrypsin hydrolysate (CH) and the ultrafiltration peptide fractions. Values are mean ± standard deviation of three determinations. Bars with different letters (a–e) on the same layout are significantly different (*p* ≤ 0.05).

**Figure 4 foods-10-01112-f004:**
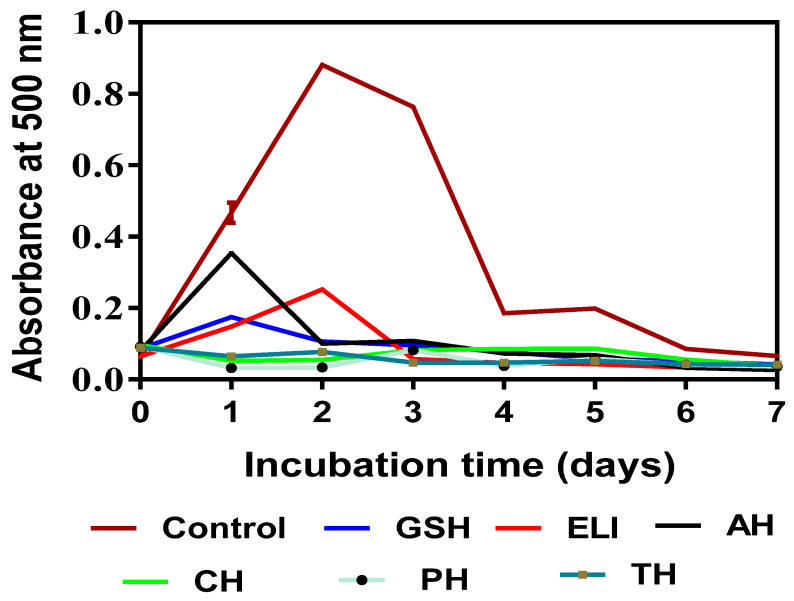
Inhibition of linoleic acid peroxidation by glutathione (GSH as control), eggplant leaf protein isolate (ELI) and enzymatic protein hydrolysates alcalase (AH), chymotrypsin (CH), pepsin (PH) and trypsin (TH).

**Figure 5 foods-10-01112-f005:**
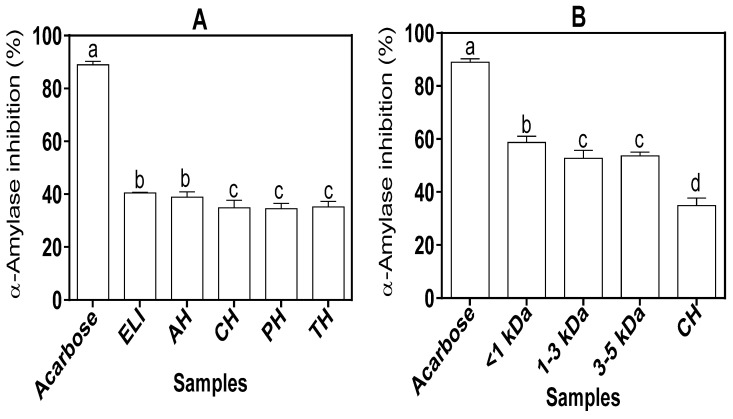
α-Amylase inhibitory activities of (**A**) acarbose (standard), eggplant leaf protein isolate (ELI) and protein hydrolysates alcalase (AH), chymotrypsin (CH), pepsin (PH),and trypsin (TH), and(**B**) acarbose (standard) and CH-ultrafiltration peptide fractions. Values are mean ± standard deviation of three determinations. Bars with different letters (a–d) on the same layout are significantly different (*p* ≤ 0.05).

**Figure 6 foods-10-01112-f006:**
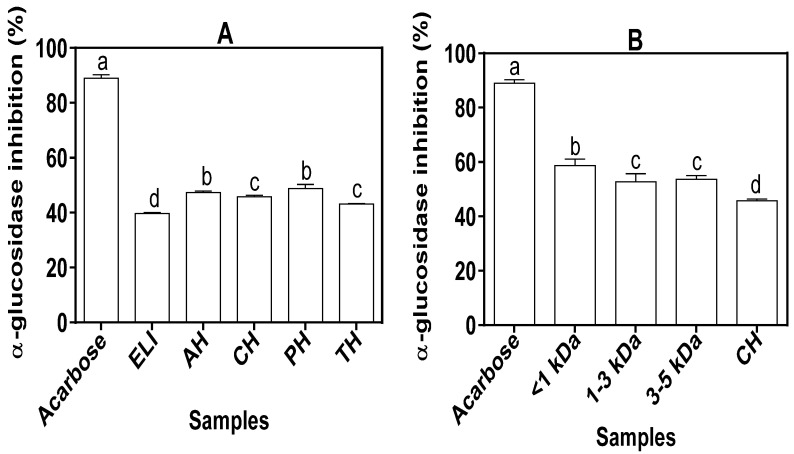
α-glucosidase inhibitory activities of (**A**) acarbose (standard), eggplant leaf protein isolate (ELI) and protein hydrolysates alcalase (AH), chymotrypsin (CH), pepsin (PH) and trypsin (TH), and (**B**) α-amylase inhibitory activities of ultrafiltration peptide fractions of CH and acarbose (standard). Values are mean ± standard deviation of three determinations. Bars with different letters (a–d) on the same layout are significantly different (*p* ≤ 0.05)

**Table 1 foods-10-01112-t001:** Amino acid composition (%) of eggplant leaf protein isolate and enzymatic protein hydrolysates *.

Amino Acid	ELI	AH	CH	PH	TH
Aspartic + asparagine	11.01	12.27	13.48	12.67	13.05
Threonine	4.55	4.80	4.38	5.12	4.40
Serine	5.56	5.41	5.30	5.85	5.20
Glutamic + glutamine	12.42	15.79	14.60	12.85	13.83
Proline	5.20	5.12	4.81	5.29	5.34
Glycine	5.15	4.73	5.23	4.93	5.17
Alanine	6.13	7.09	6.66	6.54	6.83
Cysteine	2.63	1.03	1.65	1.08	1.49
Valine	5.41	6.31	6.13	6.39	6.34
Methionine	1.86	1.41	1.46	1.44	1.46
Isoleucine	4.32	4.60	4.36	5.03	4.69
Leucine	7.78	8.65	8.17	9.07	8.11
Tyrosine	4.68	3.88	3.50	4.23	3.32
Phenylalanine	5.63	4.44	4.82	4.04	4.65
Histidine	3.35	2.65	3.38	3.02	3.21
Lysine	6.63	6.05	5.64	6.45	5.94
Arginine	5.94	4.71	4.97	4.95	5.48
Tryptophan	1.74	1.08	1.44	1.07	1.47
BCAA	17.52	19.56	18.66	20.48	19.14
HAA	45.38	43.61	43.02	44.17	43.72
AAA	12.05	9.40	9.76	9.34	9.45
PCAA	15.93	13.40	13.99	14.42	14.63
NCAA	23.44	28.05	28.08	25.52	26.89
EAA	45.96	43.86	43.29	45.84	43.60
SCAA	4.49	2.44	3.12	2.52	2.95

* ELI, eggplant leaf protein isolate; AH, alcalase hydrolysate; CH, chymotrypsin Hydrolysate; PH, pepsin hydrolysate; TH, trypsin hydrolysate; BCAA, branched-chain amino acids; HAA, hydrophobic amino acids; AAA, aromatic amino acids; PCAA, positively-charged amino acids; NCAA, negatively-charged amino acids; EAA, essential amino acids; SCAA, sulfur-containing amino acids.

**Table 2 foods-10-01112-t002:** Amino acid sequences of peptides identified in the enzymatic hydrolysates of eggplant leaf protein isolate.

Observed Mass (Da)	Calculated Mass (Da)	Peptide Sequence	Location
	Alcalase hydrolysate (AH)		

237.100	237.090	MS	f1–2, 320–321
237.100	237.123	AF	f39–40
237.100	237.123	FA	f236–257, 469–470
578.300	578.282	AYPLD	f102–106
573.300	578.283	HVWH	f383–386
578.300	578.293	LQFGGG	f400–405
578.300	578.293	ASKWS	f448–452
	Chymotrypsin hydrolysate (CH)		
305.100	305.109	DND	f286–288
375.200	375.187	AAVD	f470–473
375.200	375.187	QVE	f156–158
375.200	375.187	VEQ	f354–356
375.200	375.187	NEI	f442–444
375.200	375.199	ARE	f257–259
375.200	375.199	REA	f446–448
375.200	375.224	VKE	f17–19
375.200	375.224	GTVV	f329–332
375.200	375.224	SIVG	f119–122
375.200	375.224	TVVG	f330–333
375.200	375.224	LDK	f475–477
375.200	375.224	DKL	f160–162
391.200	391.182	EKD	f93–95
391.200	391.182	EGSV	f110–113
391.200	391.198	QPF	f209–211
391.200	391.198	WVS	f368–370
391.200	391.212	CLR	f192–194
393.200	393.180	LGCT	f170–173
393.200	393.180	GCTI	f171–174
393.200	393.191	RMS	f319–321
393.200	393.224	AFR	f39–41
393.200	393.224	FAR	f256–258
	Pepsin hydrolysate (PH)		
351.200	351.203	GFK	f12–14, 126–128
385.200	385.183	HAGT	f327–330
635.300	635.290	GGDHIH	f322–327
909.500	909.508	IAYVAYPL	f98–105
	Trypsin hydrolysate (TH)		
351.200	351.203	GFK	f12–14, 126–128
351.200	385.185	HAGT	f327–330
1018.600	1018.579	RAVFARELG	f253–261
1018.600	1018.579	HIHAGTVVGK	f325–334

**Table 3 foods-10-01112-t003:** Antioxidant activities and angiotensin-converting enzyme (ACE) inhibitory properties of eggplant leaf protein isolate, protein hydrolysates, chymotrypsin hydrolysate ultrafiltration peptide fractions and glutathione (GSH).

	DPPH Radical Scavenging Activity (EC_50_, mg/mL)	Superoxide Radical Scavenging Activity (EC_50_, mg/mL)	Hydroxyl Radical Scavenging Activity (EC_50_, mg/mL)	Metal Chelation Activity (EC_50_, mg/mL)	ACE Inhibition (IC_50_, mg/mL)
			Hydrolysates		
GSH	0.237 ± 0.019 ^c^	0.785 ± 0.008 ^b^	0.668 ± 0.023 ^e^	4.961 ± 0.016 ^a^	-
ELI	1.213 ± 0.049 ^a^	0.976 ± 0.001 ^a^	4.220 ± 0.010 ^a^	4.768 ± 0.012 ^b^	2.775 ± 0.017 ^e^
AH	0.364 ± 0.018 ^b^	0.906 ± 0.003 ^a^	3.583 ± 0.021 ^c^	4.969 ± 0.088 ^a^	2.580 ± 0.013 ^d^
CH	0.226 ± 0.001 ^c^	0.945 ± 0.002 ^a^	3.166 ± 0.009 ^d^	3.999 ± 0.601 ^e^	2.110 ± 0.069 ^b^
PH	0.264 ± 0.006 ^c^	0.958 ± 0.002 ^a^	4.230 ± 0.030 ^a^	4.292 ± 0.563 ^d^	2.382 ± 0.001 ^c^
TH	0.350 ± 0.009 ^b^	0.959 ± 0.002 ^a^	3.883 ± 0.031 ^b^	4.476 ± 0.364 ^c^	2.106 ± 0.028 ^e^
			CH Fractions		
<1 kDa	0.048 ± 0.019 ^c^	1.212 ± 0.031 ^b^	2.650 ± 0.022 ^b^	1.228 ± 0.041 ^d^	0.836 ± 0.021 ^a^
1–3 kDa	0.053 ± 0.002 ^b^	1.270 ± 0.012 ^b^	2.639 ± 0.043 ^b^	1.540 ± 0.293 ^c^	0.830 ± 0.005 ^a^
3–5 kDa	0.058 ± 0.001 ^b^	1.419 ± 0.025 ^a^	2.657 ± 0.085 ^b^	2.085 ± 0.223 ^b^	0.811 ± 0.026 ^a^
GSH	0.237 ± 0.019 ^a^	0.785 ± 0.008 ^c^	3.166 ± 0.009 ^a^	4.961 ± 0.016 ^a^	-

Values having same superscript within the same coloumn are not signficantly (*p* > 0.05) different from one another. ELI, eggplant leaf protein isolate; AH, alcalase hydrolysate; CH, chymotrypsin hydrolysate; PH, pepsin hydrolysate; TH, trypsin hydrolysate.

## Data Availability

The data will be available on request.

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
