# Peer review of "Effect of Protease Type and Peptide Size on the In Vitro Antioxidant, Antihypertensive and Anti-Diabetic Activities of Eggplant Leaf Protein Hydrolysates"

_foods, 2021, doi:10.3390/foods10051112_

Round 1

Reviewer 1 Report

This topic is important from a scientific point of view but the presentation of the paper needs to improve.

There were many spelling mistakes and unexplained acronyms. To highlight my point, I am listing a few that I found in the paper.

Line 18.  Hydrolydates should be “hydrolysates”.

Lines 19-20. Ambiguous sentence

Line 37. What is meant by Cryptides?

Lines 57-58. This sentence does not make sense.

Lines 69-71. These two sentences do not make sense. You seem to use the words “proteins” and “protein hydrolysates” interchangeably, which is not correct.

Section 2, Materials and Methods. The various assay methods that you followed from previously published papers were given as the method followed by a reference number (??).

For example, Eggplant leaf protein isolate (ELI) was obtained by a modified method of [28].

It should be written as “Eggplant leaf protein isolate (ELI) was obtained by a modified method of Malomo, et al, (2014) [28]”.

Line 99. Why was this sentence highlighted?

Line 196. What is GSH? Spell out in full.

Line 238-240. This sentence is not grammatically correct.

Line 262. Explain briefly what acarbose is (an antidiabetic drug used to treat Type 2 diabetes).

Section 2.10 Statistical Analysis. The plural of analysis is “Analyses”.  What is the reference for the statistical package that you used?

Section 3.1. How can you get different protein contents if you started off with the same Eggplant Leaf protein Isolate (ELI) for hydrolyses by different proteases?

Could it be due to the added proteases contributing to the total protein content?

Although you have given the reason for the differences in the protein content in the Discussions, you should state them here under Section 3.1, when you are discussing the results for DH and protein content.

Lines 290-291. This sentence does not make sense.

Section 3.2. Amino acid composition.

Lines 323-326. Again, my question is how can you get differences in the amino acid composition of different hydrolysates, if you started off with the same EFI?

Lines 323- 326. These two sentences imply that somehow the amino acid composition changed during the hydrolyses process.

Line 327. Table 1. If the samples were completely hydrolysed for amino acid analyses, you should not get any difference between the unhydrolyzed ELI and its partially hydrolysed products.

Section 3.6.

Table 3. Should also include what GHS is, in the caption.

Table 4. What is the control used? It should be stated in the caption. Also include what GHS is, in the caption.

Figure 5. Include what acarbose is, in the caption.

Figure 6. Include what acarbose is, in the caption.

Section 4. Discussion

Line 439. This sentence should be supported by an appropriate reference. Otherwise, it will be just conjecture.

Line 441. This sentence should begin with “Paiva, et al, (2017) (40) reported……..”

Line 515-516. Why is this sentence highlighted?

Author Response

REVIEW 1

Line 18.  Hydrolydates should be “hydrolysates”.

  • The spelling mistake has been effected in line 18

Lines 19-20. Ambiguous sentence

  • the sentence explained the rationale behind the selection of chymotrypsin hydrolysate (CH) for fractionation, because for its overall antioxidant properties when compared with other protein hydrolysates (lines 19-20)

Line 37. What is meant by Cryptides?

  • cryptides are bioactive peptides

Lines 57-58. This sentence does not make sense.

  • the sentence has been reformatted (lines 59-60)

Lines 69-71. These two sentences do not make sense. You seem to use the words “proteins” and “protein hydrolysates” interchangeably, which is not correct.

  • the sentence has been modified (71-73)

Section 2, Materials and Methods. The various assay methods that you followed from previously published papers were given as the method followed by a reference number (??).

For example, Eggplant leaf protein isolate (ELI) was obtained by a modified method of [28].

It should be written as “Eggplant leaf protein isolate (ELI) was obtained by a modified method of Malomo, et al, (2014) [28]”.

  • The citing of the references in the text have been modified

Line 99. Why was this sentence highlighted?

  • the highlight was done in error. I have effected the correction (lines 99-100)

Line 196. What is GSH? Spell out in full.

  • GSH is glutathione and this has been effected in the document (line 200)

Line 238-240. This sentence is not grammatically correct.

  • the sentences have been edited and corrected appropriately (lines 239-247)

Line 262. Explain briefly what acarbose

  • acarbose is an antidiabetic drug used to treat Type 2 diabetes.

Section 2.10 Statistical Analysis. The plural of analysis is “Analyses”.  What is the reference for the statistical package that you used?

  • the reference has been inserted (294-296)

Section 3.1. How can you get different protein contents if you started off with the same Eggplant Leaf protein Isolate (ELI) for hydrolyses by different proteases?

Could it be due to the added proteases contributing to the total protein content?

Although you have given the reason for the differences in the protein content in the Discussions, you should state them here under Section 3.1, when you are discussing the results for DH and protein content.

  • Discussion is not allowed in the “Results” section, hence we used the “Discussion” section.

Lines 290-291. This sentence does not make sense.

  • the sentence has been edited and corrected (294-296)

Section 3.2. Amino acid composition.

Lines 323-326. Again, my question is how can you get differences in the amino acid composition of different hydrolysates, if you started off with the same EFI?

Lines 323- 326. These two sentences imply that somehow the amino acid composition changed during the hydrolyses process.

Line 327. Table 1. If the samples were completely hydrolysed for amino acid analyses, you should not get any difference between the unhydrolyzed ELI and its partially hydrolysed products.

  • the amino acid composition of the hydrolysates can be different from the unhydrolyzed protein because not all the protein is converted to peptides during enzyme hydrolysis. Note that after enzyme hydrolysis, the digest is centrifuged to removed undigested proteins as the precipitate while the supernatant is used as the hydrolysate. The hydrolysed samples were hydrolysed by enzymes, partially and have undergone post hydrolysis processes, hence the difference in the amino acid composition. Also, due to differences in enzyme specificity, the peptides will differ in amino acid composition depending on the protease used.

Section 3.6.

Table 3. Should also include what GHS is, in the caption.

  • GSH has been inserted in the caption (lines 367-368)

Table 4. What is the control used? It should be stated in the caption. Also include what GHS is, in the caption.

  • There is no Table 4 in this work. If the question is about Table 3, then the standard used was glutathione (GSH) (lines 367-368)

Figure 5. Include what acarbose is, in the caption.

  • this has been included (line 415)

Figure 6. Include what acarbose is, in the caption.

  • this has been included (line 427)

Section 4. Discussion

Line 439. This sentence should be supported by an appropriate reference. Otherwise, it will be just conjecture.

  • Reference added 

Line 441. This sentence should begin with “Paiva, et al, (2017) (40) reported……..”

  • This has been effected (line 437)

Line 515-516. Why is this sentence highlighted?

  • this was done in error. It has been corrected (lines 515-516)

Reviewer 2 Report

This manuscript assesses the effect of protease type and peptide size on the antioxidant, antihypertensive and anti-diabetic activities of eggplant leaf protein hydrolysates. The introduction clearly explains what the authors want to study, but, although the idea of using eggplant leaves as a substrate to obtain proteins is original, I have some concerns about the study.

The aim of the study was to determine the effect of protease type and peptide size on the antioxidant properties and inhibitions of the activities of ACE, α-amylase and α-glucosidase. However, the main conclusion of the study is that the peptide composition and in vitro bioactive properties of protein hydrolysates produced from eggplant leaf protein isolate were dependent on the enzyme type, which is too obvious, and no evidence is shown to assess the effect of peptide size on the biological activities.

Abstract should include a conclusion paragraph.

Authors hydrolyzed ELI with different enzymes and then selected CH to obtained peptide fractions. The criteria for the selection of CH is not clear as in some parts of the text indicates that is the antioxidant activity and in other parts the antioxidant activity and the ACE inhibitory activity. Moreover, the higher antioxidant and ACE inhibitory activity of CH is not clear attending to the presented data.

Authors indicate that leaf proteins can serve as a better alternative to the expensive animal and seed-based proteins. However, they used acetone for leaf power preparation. An alternative method should be sought to obtain a food grade final product.

The sections materials and methods is too long.

What are the proteins from eggplant leaf? Have the authors made any characterization? At least a SDS-PAGE should be included.

Table 2 indicates location of the peptide, but what does that mean? What protein sequences have been used to match mass identified by mass spectrometry? It seems that identification of many peptides is missing. What does the asterisk indicate?

Table 3. What do the superscript letters mean?

The method description in section 2.3.1 is not clear. The absorbance is the final readout, but no absorbance values are included in the equation.

Section 2.10. The p value considered significant should be specify.

Figure 1 and 4 should include error bars.

The discussion is extremely long.

Line 171: was subsequently was incubated

Line 188: sample was determined sample by

Line 291: signficsntly

Line 296-297 should be on the same line

Line 391: Hydrolysates should be removed

Format of the references should be carefully checked.

Author Response

The aim of the study was to determine the effect of protease type and peptide size on the antioxidant properties and inhibitions of the activities of ACE, α-amylase and α-glucosidase. However, the main conclusion of the study is that the peptide composition and in vitro bioactive properties of protein hydrolysates produced from eggplant leaf protein isolate were dependent on the enzyme type, which is too obvious, and no evidence is shown to assess the effect of peptide size on the biological activities.

  • thank you. The present work only deals with the in-vitro activities. Testing of the peptide for biological activities will be the focus next time

Abstract should include a conclusion paragraph.

  • the conclusion has been included in the abstract (lines 29-31)

Authors hydrolyzed ELI with different enzymes and then selected CH to obtained peptide fractions. The criteria for the selection of CH is not clear as in some parts of the text indicates that is the antioxidant activity and in other parts the antioxidant activity and the ACE inhibitory activity. Moreover, the higher antioxidant and ACE inhibitory activity of CH is not clear attending to the presented data.

  • Thank you. The CH was selected based on the overall antioxidant activities. The word overall was used because, out of the five antioxidant assays, CH was more potent in four out of five, hence, the decision to further fractionate the CH.

Authors indicate that leaf proteins can serve as a better alternative to the expensive animal and seed-based proteins. However, they used acetone for leaf power preparation. An alternative method should be sought to obtain a food grade final product.

  • Thank you. We shall consider obtaining a food grade product by using non-organic solvent next time.

The sections materials and methods is too long.

  • The Journal style demands that we explain in details, the procedures for analyses and to allow for repeatability of the work

What are the proteins from eggplant leaf? Have the authors made any characterization? At least a SDS-PAGE should be included.

  • Ribulose-1,5-biphosphase carboxylase-oxygenase (RuBisco) is the major eggplant leaf protein and has been characterized. The SDS-PAGE has been done and published in the International Journal of Food properties (2020, 23, 955-970).

Table 2 indicates location of the peptide, but what does that mean?

  • the location, as indicated in Table 2, shows the position of the peptide sequence within the parent protein (RuBisco) primary structure.

What protein sequences have been used to match mass identified by mass spectrometry?

  • RuBisco

It seems that identification of many peptides is missing.

  • based on the mass tolerance use (+0.025), no peptide is missing.

What does the asterisk indicate?

  • the asterik was in error. It has been deleted from the Table.

Table 3. What do the superscript letters mean?

  • the superscript represent significant level and this has been stated.

The method description in section 2.3.1 is not clear. The absorbance is the final readout, but no absorbance values are included in the equation.

  • the absorbance value is what is imputed in the equation

Section 2.10. The p value considered significant should be specify.

  • this has been inserted (line 294-296)

Figure 1 and 4 should include error bars.

  • the error bars have been included. Some may not be visible due to small errors among the numbers (lines 309-310; 396-397)

The discussion is extremely long.

  • The discussion needed to be long to be able to explain everything in the work to details.

Line 171: was subsequently was incubated

  • this has been corrected accordingly (line 174)

Line 188: sample was determined sample by

  • this has been appropriately corrected (line 189)

Line 291: signficsntly

  • the spelling error has been done (295)

Line 296-297 should be on the same line

  • this has been done (297-298)

Line 391: Hydrolysates should be removed

  • it has been effected (line 390)

Format of the references should be carefully checked

  • this has been checked

Reviewer 3 Report

Authors of the publication "Effect of Protease Type and Peptide Size on the Antioxidant, Antihypertensive and Anti-diabetic Activities of Eggplant Leaf Protein Hydrolysates " they did the hydrolysis of Solanum macrocarpon proteins and assessed the biological activity of the hydrolysates. The work is interesting and scientifically substantiated. Explorations of natural compounds that can be used in the treatment of diabetes and hypertension are very important. 
However, the reviewer has some comments:

The authors refer to the previously published method for the determination of amino acids by HPLC (reference 31). There is no mention of HPLC in the publication cited by the authors. How will the authors explain it?

Figure 4 - In which units was the absorbance measured? @?

The abbreviation GSH should be explained on first use. Also in the description of the figures.

In each case, it should be clearly defined what was a positive and what was a negative control.

I would be cautious in conclusions regarding the possibility of using hydrolysates as drugs. Their activity is much less than that of acarbose. Moreover, the reviewer did not find data on the activity of the negative control in the graphs.

Minor:

Please standardize the font format in figures 5 and 6.
Write Latin names of plants in italics, e.g. line 153.

Author Response

The authors refer to the previously published method for the determination of amino acids by HPLC (reference 31). There is no mention of HPLC in the publication cited by the authors. How will the authors explain it?

  • Thank you. These are the original citations of the methods are  -acid hydrolysis (Bidlingmeyer  et al., 1984), Performic acid oxidation (Gehrke  et al., 1985) and alkaline hydrolysis Landryl  et al. (1982). However, Girgih et al. (2011), which was cited in this work integrated the three methods to determine amino acids of hemp seeds and that is why it was cited to shorten the already long methods section.

Figure 4 - In which units was the absorbance measured? @?

  • Usually, absorbance does not have a true unit. @ has been replaced with ‘at’.

The abbreviation GSH should be explained on first use. Also in the description of the figures.

  • GSH has now been defined on first use and on the captions.

In each case, it should be clearly defined what was a positive and what was a negative control.

  • Only the positive control is reported in all the assays because the negative control (Water or buffer) has been subtracted from the samples.

I would be cautious in conclusions regarding the possibility of using hydrolysates as drugs. Their activity is much less than that of acarbose. Moreover, the reviewer did not find data on the activity of the negative control in the graphs.

  • the values of the negative control has been subtracted from the samples and that of the standard, that is why the negative control did not feature on the graphs

Minor:

Please standardize the font format in figures 5 and 6.

  • this has been adjusted
    Write Latin names of plants in italics, e.g. line 153.
  • this has been corrected (line 156)

Round 2

Reviewer 1 Report

  1. You have corrected some of the mistakes in the revised version.
  2. However, the Lowry method that you used to measure protein is not a standard method and is affected by phenolics and reducing substances found in the leaf protein isolates and even K+, Mg2+, NH4+, EDTA, Tris-HCl can interfere with the method.  
  3. What is the control in Figure 4? It should be stated in the Figure caption.

Author Response

Review 1

  1. The Lowry method that you used to measure protein is not a standard method and is affected by phenolics and reducing substances found in the leaf protein isolates and even K+, Mg2+, NH4+, EDTA, Tris-HCl can interfere with the method.  

Response: The Lowry method is a standard method of protein determination and is unaffected by EDTA, sucrose or salts (Markwell et al. 1978. Anal. Chem. 87, 206-210). The leaf powder was treated with two consecutive rounds of acetone extraction to remove the polyphenolic compounds (section 2.1), therefore level of interference is inconsequential to the reported values. Please note that all standard protein measurement methods suffer from one form of interference or the other but these are very minor and the protocol remains valid as long as same method is used for all the samples.

  1. What is the control in Figure 4? It should be stated in the Figure caption.

Response: The control is glutathione (GSH) and has been indicated in the caption (line 416)

Reviewer 2 Report

  • If the present work only deals with the in-vitro activities, then the “in vitro” expression should be stated in the title. Moreover, if authors do not show any evidence about the effect of peptide size on the biological activities, the title should be “Effect of protease type on the in vitro antioxidant, antihypertensive and anti-diabetic activities of eggplant leaf protein hydrolysates”.
  • The equation in section 2.3.1 is still not clear. The number of free amino groups at X min of enzymatic hydrolysis is included in the equation. How is this calculated?
  • The discussion is extremely long.

Author Response

  • Review 2

    • If the present work only deals with the in-vitro activities, then the “in vitro” expression should be stated in the title. Moreover, if authors do not show any evidence about the effect of peptide size on the biological activities, the title should be “Effect of protease type on the in vitro antioxidant, antihypertensive and anti-diabetic activities of eggplant leaf protein hydrolysates”.

    The title of the work has been adjusted to include in vitro as suggested (line 2)

    • The equation in section 2.3.1 is still not clear. The number of free amino groups at X min of enzymatic hydrolysis is included in the equation. How is this calculated?

    (NH2)x=Amount of free amino groups in the supernatant after x min of hydrolysis (0.5, 1.0, 1.5, 2.0, 2.5, 3.0, 3.5, 4.0 h as shown in Figure 1). In this experiment, an aliquot of the protein digest was taken at 30 min (0.5 h) intervals for analysis and this represents the ‘x’ min in the equation. The text has been amended to indicate sampling of the protein digests at every 30 min (x values).

    • The discussion is extremely long

    Response: The discussion is long because of the large amount of data presented. However, some parts of the discussion have been deleted. Further reductions in length will compromise the ability of readers to understand and use the results.

Reviewer 3 Report

The authors did not clarify my doubts about the determination of amino acids by HPLC (section 2.5). This section deals with HPLC analysis and the cited literature (31) does not provide this information. How to the authors determinated amino acids?
Is there any reason why the authors do not want to provide a methodology for the determination of amino acids? I am not convinced by the authors' arguments. The tests must be reproducible and verifiable. Without a solid methodological part, reviewing the results makes dubious sense. Sorry, but I am a chromatographist and based on this work, I would not know how to determine amino acids.

Author Response

  • Review 3

    The authors did not clarify my doubts about the determination of amino acids by HPLC (section 2.5). This section deals with HPLC analysis and the cited literature (31) does not provide this information. How to the authors determinated amino acids?
    Is there any reason why the authors do not want to provide a methodology for the determination of amino acids? I am not convinced by the authors' arguments. The tests must be reproducible and verifiable. Without a solid methodological part, reviewing the results makes dubious sense. Sorry, but I am a chromatographist and based on this work, I would not know how to determine amino acids.

    • An expanded description of the amino acid analysis has now been provided.

This manuscript is a resubmission of an earlier submission. The following is a list of the peer review reports and author responses from that submission.